# Do whispering minds tingle alike? Exploring the relationship between ASMR-sensitivity, trait-ASMR, and trigger preference

Joanna M. H. Greer[1¤a]*, Colin J. Hamilton[1], Daniela Beckelhymer[2¤a], Emily Thompson[2¤b], Carin Perilloux[2]

1 Department of Psychology, Northumbria University, Newcastle-upon-Tyne, United Kingdom,
2 Department of Psychology, Southwestern University, Georgetown, Texas, United States of America

¤a Current address: Department of Mathematics, University of Minnesota-Twin Cities, Minneapolis, Minnesota, United States of America
¤b Current address: Department of Mathematics, Southern Methodist University, Dallas, Texas, United States of America
* Joanna.greer@northumbria.ac.uk

## Abstract

The Autonomous Sensory Meridian Response (ASMR) is described as a pleasant tingling sensation originating in the head and neck in response to specific audio-visual stimuli (triggers) with individuals self-classifying as ASMR-responders based on their experience of the phenomenon. There is great variability in the types of triggers that elicit ASMR across individuals, with previous attempts to categorise triggers encompassing multiple affective components across sensory domains. More recently, research indicates ASMR should be considered a personality construct, with traits falling on a spectrum regardless of ASMR-sensitivity. The current study recruited an extensive sample of ASMR viewers (n = 16,679) to investigate character-istics of trait-ASMR using psychometric properties of the ASMR-15 in self-classified ASMR-responders who either experienced 'authentic' ASMR where tingles originated in the head and neck or 'non-authentic' ASMR, where tingles originate in other bodily regions. Secondly, we explored group differences in trigger intensity by trait-ASMR and ASMR-sensitivity. K-means cluster analysis identified tripartite trait-ASMR categorisation (High/ Medium/ Low). These cluster analysis groups showed distinct profiles across the ASMR-15 subscales evidenced with medium effect sizes, whereas the difference based on authentic vs. non-authentic ASMR experience was a small effect. To identify trigger categorisation, principal component analyses revealed four key categories: *Roleplay, Watching, Interpersonal care*, and *Tactile*. Group differ-ences by trait-ASMR grouping across the trigger categorisation revealed small to medium effects, whereas effect sizes by ASMR-sensitivity were predominantly small to negligible. This suggests we should conceptualize individual differences in ASMR experience as a trait on a spectrum rather than a dichotomous categorical variable

**Data availability statement:** All data are available from the Open Science Framework: https://osf.io/j34d8/.

**Funding:** The author(s) received no specific funding for this work.

**Competing interests:** The authors have declared that no competing interests exist.

since only trait classification accounted for individual differences in ASMR-propensity and trigger categorisation.

---

## Introduction

The Autonomous Sensory Meridian Response (ASMR) is described as a tingling, static-like sensation in response to gentle and low frequency audio, visual, and tactile stimuli, that originates across the scalp, neck, spreading down the spine [1]. Since its appearance on online forums in the early 21st century, the interest in ASMR has escalated at an unprecedented rate, with content seeking to trigger ASMR uploaded to multiple platforms including YouTube, TikTok, and Twitch. These videos often involve an actor (known as an ASMRtist) performing a scenario such as simulating haircutting, medical appointments, and other forms of personal care directly into the camera and which is typically accompanied with a gentle whispering auditory. While the conventional definition of ASMR refers to tingles originating in core areas of the head and neck, the sensation can be experienced across the body including the arms and legs [2]. It is accompanied with feelings of relaxation or sleepiness [3–5] and those who seek out ASMR content report that it helps with psychological states such as depression and anxiety [1]. The tingling sensation, though intentionally stimulated, occurs involuntarily [6] and has been compared to the sensations of misophonia, frisson, synesthesia, and paresthesia [1,7,8]. However, despite comparisons with these affective responses, ASMR is considered distinct due to its idiosyncratic intensity and duration, with tingles spreading to peripheral body regions with increasing intensity [9].

The emotional reaction in response to ASMR content can be considered as a temporary state (state-ASMR) that is reflective of individuals' underlying propensity to experience ASMR (trait-ASMR). While the triggering stimuli that evoke the ASMR sensation are highly subjective to the individual [10,11], ASMR experiencers commonly cite social (e.g., personal attention), tactile (e.g., hair brushing, gentle touches) and other gentle modalities (e.g., tapping, whispering, visually satisfying images, nature sounds) as eliciting ASMR [1,4,12]. Two studies have aimed to develop ASMR checklists that capture most frequently endorsed triggers that are broad enough to encompass the multisensory aspect of the ASMR experience. Fredborg et al. [13] asked ASMR-responders to rate intensity of tingles in response to fourteen known triggering stimuli, finding *Whispering, Haircut simulation*, and *Tapping sounds* the most popular. The lowest ratings were in response to *Chewing sounds* and *Watching others cook*. Using Principal Component Analyses (PCA), the authors revealed a five-factor model, with trigger categories labelled as *Watching, Touching, Repetitive Sounds, Simulations*, and *Mouth Sounds*. The latter was of particular interest as both *Whispering* and *Chewing sounds* loaded strongly onto this factor despite polarizing intensity ratings from the participants. More recently, Poerio et al. [14] created a 37-item ASMR Trigger Checklist (ATC). ASMR-responders were asked to endorse and rate intensity of tingles to a series of triggers, as well as provide feedback as to how well they aligned with four categories: *Vocal-auditory* (VA), *Non-vocal-auditory*

(NVA), *Visual* (Vis), and *Tactile/ Interpersonal care* (T/IPC). Overall, participants endorsed an average of 6.76 triggers as eliciting ASMR. The most endorsed triggers represented four trigger categories (e.g., soft-speaking (VA – 94%), brushing sounds (NVA – 93%), delicate hand movements (Vis – 92%), physical touch on body (T/IPC – 98%), thus emphasising ASMR as a multi-sensory experience, with triggers encompassing multiple affective modalities.

The psycho-social emotional characteristics of the ASMR phenomenon are tentatively suggested as an evolutionary adaptive trait, with the pleasurable response considered a 'form of social grooming' [4(p.14), also see 15] and likened to social interactions such as allogrooming observed in non-human primates [16]. This is further supported with a comparison of ASMR with tactile experiences associated with affective touch (AT) such as stroking or caressing [17,18]. The pleasurable response to AT is functionally associated with activity along unmyelinated C-tactile afferent nerve fibres, which are associated with subjective ratings of pleasure and positive emotional effects of physical closeness [19–21]. Linking this with ASMR content, a recent study analysed 2,663 ASMR-related videos uploaded to YouTube, identifying three overarching features: diverse social connections, a relaxing sense of bodily intimacy, and sensory-rich observational activities [22]. Fifty nine percent of videos included a minimum of one type of touch interaction, and ~29% included scenarios where the ASMRtist reached toward the camera and simulated a physical interaction with the viewers' face or body. While watching ASMR videos does not involve physical tactile stimulation for the viewer, the visual-auditory element may provide enough relevant cues to stimulate the same affective responses associated with AT. Both are accompanied by physiological changes indicative of a relaxed state such as decreased heart rate [4,23] as well as a more effective stress regulation response [24,25]. Furthermore, functional Magnetic Resonance Imaging (fMRI) identifies activity in brain regions functionally associated with reward and emotion including the insula, the medial prefrontal cortex, and the dorsal anterior cingulate cortex in response to both ASMR exposure and AT [15,26,27]. Importantly, qualitative perspectives from ASMR-users suggest that it is the psychological impact not the somatic experience of ASMR that is more meaningful [28].

The aforementioned research is characteristic of much of the empirical literature which typically takes a dichotomous perspective of ASMR-sensitivity by comparing those who self-report as experiencing ASMR with those who do not [4]. Importantly, this classification is attributed to a physiological response which is indicative of state-ASMR. As such, a binary perspective masks nuances in the ASMR experience due to individual differences in both preference of trigger type and brain activity in response to different ASMR stimuli [2,5,13]. Furthermore, within a self-categorization process false positives are observed, with some participants self-classifying as ASMR-sensitive despite tingles originating in more peripheral bodily regions, and which could be the result of expectancy effects [29] (see [30] for a response). However, psychological and therapeutic benefits of ASMR exposure have been observed in both ASMR-sensitive and non-sensitive individuals [5,24], while in a separate study, no therapeutic benefit was observed in some ASMR-sensitive individuals despite their tingles originating in the head and/ or neck [31]. As such, ASMR-propensity should not be considered binary (i.e., experiencer vs. non-experiencer). Rather it is more plausible that ASMR-propensity is considered a personality construct which lies on a continuum (trait-ASMR), with individuals' trait-ASMR characteristics more informative regarding the benefits of ASMR exposure rather than on categorisation based on a self-reported physiological response. Most trait individual differences are more dimensional than can be categorised by a binary grouping, therefore restricting ASMR-propensity in this manner also potentially misses these nuances [32,33].

Recently, two measures have been developed that enable a more nuanced perspective of ASMR-propensity. The ASMR Experience Questionnaire (AEQ; [2]) investigates the physiological aspects of ASMR experience via body location and intensity. Using k-means cluster analysis, Swart et al. [2] identified five key categories of ASMR-responders based on ASMR-propensity and intensity, as well as affective responses to ASMR-exposure. Their data-driven approach revealed an ASMR-experience profile ranging from strong through to weak, along with a false positive category. This emphasises that, rather classifying ASMR-responders by binary self-reported categories of experiencer/ non-experiencer, ASMR experience should be considered as a spectrum, and which may differ depending on preferences for ASMR modality (e.g., visual vs. auditory stimuli) [5].

A second measure, the ASMR-15 [8] is a 15-item psychometric tool that measures trait-ASMR across 4 subscales: Altered Consciousness (shifts in consciousness and mental functioning), Sensation (physical feelings and tingles), Relaxation (feeling calm, sleepy, and relaxed), and Affect (feelings of pleasure, bliss, and euphoria). As a psychometrically validated scale, it currently appears the most informative tool to delineate variability in trait-ASMR as well identifying individual differences in the ASMR experience. For example, and as might be expected, higher ASMR-15 scores (indicative of greater trait-ASMR) are reported in ASMR-sensitive participants compared to non-experiencers [24,32]. However, Zielinski-Nicolson et al. [33] observed significant positive relationships between trait-ASMR, based on ASMR-15 scores, and various characteristics such as roleplaying, creativity, and schizotypal traits. Finally, Lohaus et al. [12] found differences in overall ASMR-15 scores in the Relax and Affect subscales depending on content of the ASMR video.

As outlined above, binary self-classification an either an experiencer or non-experiencer based on ASMR-sensitivity does not accurately reflect individuals' trait-ASMR. Similarly, categorising individual triggers is problematic due to the subjective nature of ASMR viewers' experiences and preferences. Thus, the aim of the current study was threefold: a) examine the profile of trait-ASMR within the ASMR-viewing community from a data-driven perspective by using the psychometric properties of the ASMR-15 (High/ Medium/ Low trait-ASMR) compared to self-reported ASMR-sensitivity (authentic – where tingles originate in the head and/ or neck vs. non-authentic – where tingles originated elsewhere in the body); b) investigate the categorisation of triggers using PCA; and c) explore differences in trigger categorisation depending on trait-ASMR and ASMR-sensitivity grouping. The current study took a data-driven approach employing PCA to identify whether a large sample of trigger stimuli mapped onto the categories identified by Fredborg et al. [13] and Poerio et al. [14]. In doing so, the study aimed to demonstrate the extent to which triggers are discrete in their categorisation or are multi-dimensional by cross-loading onto other categories, and how these differ by trait-ASMR and ASMR-sensitivity. To pre-empt the results, as the data were collected from a very large sample drawn from the ASMR-viewing community, we expected the participants to predominantly report authentic ASMR and have greater trait-ASMR characteristics with higher scores on the ASMR-15 compared to non-authentic ASMR responders. As the PCA was exploratory, no predictions were made regarding categorisation. However, it was predicted that individuals High in trait-ASMR would endorse more triggers and give greater intensity ratings compared to those Low in trait-ASMR.

## Materials and methods

### Design

The study employed a non-experimental factorial design with trait-ASMR grouping based on k-means cluster analysis, and ASMR-sensitivity grouping based on the origin of tingles as the group variables. The dependent variables (DV) were trigger endorsement and trigger intensity rating scores. Principal component analysis (PCA) was conducted to identify trigger categories. Group comparisons based on trait-ASMR and ASMR-sensitivity were conducted, with factor loading scores as the DVs.

### Participants

An a priori G*power analysis [34] identified 366 participants were required to achieve 80% power for detecting a small effect [35] with a significance criterion of α.05. Participants (n = 29,033) were recruited from ASMR communities on Reddit, Facebook, and Instagram, and via the YouTube channel of the ASMRtist GentleWhispering, who has an audience of over one million subscribers. Post data cleaning (non-completed surveys including: missing age, responding 'No' to whether they experienced ASMR, non-completed questionnaires, and missed attention checks; n = 12,354), our final sample consisted of 16,679 participants aged 18–78 years (12,829 women, 3,356 men, 403 non-binary, 44 other gender, and 47 who preferred not to say (PNS) with an age range of 18–78 years (M = 26.62 years; SD = 8.16). Participants reported their ethnicity as White (83.6%), Hispanic or Latino (11.6%), Asian (4.8%), Black or African American (1.9%), American Indian or Alaska Native (1.3%), Native Hawaiian or Other Pacific Islander (0.4%), or Other (4.0%) (participants could select as

many ethnicities as applicable). Regarding relationship status, 44.4% of participants were single, 51.3% in a committed (closed) relationship, 2.5% in an open relationship, and 1.8% PNS. Of the final sample, 15,926 (95.5%) classed themselves as ASMR-experiencers, 752 (4.5%) unsure if they experienced ASMR, and 1 PNS (see Table 1).

## Materials

**ASMR-15.** Trait-ASMR was measured using the ASMR-15 [8] which consists of 15 items and four subscales: Altered Consciousness (AC; 4-items), Sensation (5-items), Relaxation (Relax; 3-items), and Affect (3-items). Example items include 'It feels like a different state of mind' and 'The sensation feels like a wave of energy'. Participants rated each item from 1 ('completely untrue for me') to 5 ('completely true for me'). Mean scores were calculated for each subscale, and a mean for the overall scale, with greater scores indicative of greater trait-ASMR. All four subscales exhibited acceptable reliability in our sample (αAltered Consciousness = .77, αSensation = .77, αRelaxation = .75, αAffect = .73), as did the overall ASMR-15 scale (α = .81).

**Tingle location and intensity.** To identify the location of tingles in response to ASMR exposure, participants were presented with an interactive bodymap, divided into 14 different locations distributed across the whole body (see [2]). Participants were instructed to identify where their tingles originated by clicking on a maximum of three locations. Next, participants had to identify where they felt the greatest intensity of tingles by clicking the relevant location on the bodymap, with a maximum of three locations.

**Trigger endorsement and tingle intensity.** To identify ASMR trigger preferences, participants were provided a list of 58 ASMR trigger scenarios such as applying make-up, personal care, tapping, mouth sounds, and various role play simulations (see S4 Table). The triggers were selected based on an extensive search of online ASMR video content, as well as ASMR forums where trigger preferences were discussed. Participants were asked to check the triggers they preferred and to leave unchecked any triggers they did not like. Participants could endorse as many triggers as applicable, and the sum of endorsed triggers was calculated. Participants were also asked to rate how much they enjoyed each trigger they endorsed on a scale of 1 (enjoy a little) to 3 (one of my favorites). Greater scores reflected greater enjoyment. Non-response to any trigger scored zero. The mean trigger rating score was calculated by summing the trigger ratings and dividing by 58.

## Procedure

Participants accessed the survey via a link to Qualtrics posted in online ASMR-specialist forums on Reddit, Facebook, Instagram, and the YouTube channel of the ASMRtist, GentleWhispering. Informed written consent required participants to respond 'Yes' to three statements: 1) 'You have read and understood the statements above' (this related to the participant information sheet), 2) 'You are willing to participate in the study as described above', 3) 'You are at least 18 years old'. Any participant answering 'No' to any question was exited from the survey. Participants provided demographic details including age, gender, ethnicity, and relationship status. Participants were then asked to respond whether they believed they experienced ASMR or not. Participants responding either 'Yes', 'Unsure', or 'PNS' were directed to the questions relating

Table 1. Participants' demographic details of gender, age (SDs in parentheses), and ASMR-experience self-classification.

| Demographics | Gender | Woman | Man | Non-Binary | Other | PNS | Total |
|---|---|---|---|---|---|---|---|
| | n | 12,829 | 3,356 | 403 | 44 | 47 | 16,679 |
| | Age | 26.39 (7.94) | 27.87 (8.96) | 23.93 (6.19) | 26.52 (8.28) | 23.72 (6.85) | |
| ASMR- experience self-classification | Yes | 12264 | 3198 | 39 | 40 | 45 | 15,926 |
| | Unsure | 564 | 158 | 24 | 4 | 2 | 752 |
| | PNS | 1 | n/a | n/a | n/a | n/a | 1 |

to the ASMR-15, the bodymap for tingle locations and intensity, trigger endorsement and trigger intensity ratings. The participants who answered 'No' to experiencing ASMR skipped these sections and completed a series of questionnaires not reported here. Four attention checks were interspersed throughout the survey. Participants who missed any of the four attention checks were removed from the final sample. The survey was anonymous therefore it was not possible to identify individual participants from any of their data. The study received ethical approval from the Institutional Review Board at Southwestern University (ref: SP19_20). Data were collected from 10th – 17th June 2019.

### Statistical analyses

Data were analysed using IBM SPSS v29 statistical software. Participants were categorised as either experiencing authentic-ASMR (aASMR) if their tingles originated in the head and/ or neck, or non-authentic-ASMR (nASMR) if their tingles originated in any other part of the body [1]. K-means cluster analyses were conducted to identify trait-ASMR group allocation (High/ Medium/ Low) based on scores from the subscales from the ASMR-15. Two PCAs were conducted on the mean ratings of trigger preference scores to identify the classification of triggers. A series of univariate analyses of variance (ANOVA) and multivariate analyses of variance (MANOVA) were conducted to explore trait-ASMR group differences (High/ Medium/ Low) and ASMR-sensitivity (aASMR/ nASMR) group differences on ASMR-15 scores, trigger endorsement, mean overall trigger ratings, and trigger classification ratings. Due to the size of the sample, it was expected that most analyses would be significant ($p < .05$) therefore conclusions were drawn predominantly on effect sizes (Cohen's d): small effect ~0.20, medium effect ~0.50, large effect ~0.80, very large effect ≥ 1.00 [35]. Due to the large sample size, multiple comparisons were conducted using either Bonferroni corrected pairwise comparisons which provides a more conservative statistic [36] or the Scheffé test which is more sensitive to large sample sizes [37]. The most conservative p value is reported. Full inferential statistics not reported here can be found in the Supplementary Materials. The dataset for the study can be accessed via: https://osf.io/j34d8/.

## Results

### ASMR-sensitivity and trait-ASMR cluster group allocation

Participants were allocated to either authentic-ASMR (aASMR) or non-authentic ASMR (nASMR) grouping based on the self-reported location of their tingles as identified via the bodymap. Participants who identified their tingles originated in the head and/ or neck were allocated to the aASMR group (n = 15,469), and participants whose tingles originated in any other part of the body were allocated to the nASMR group (n = 1,210). An independent samples t-test confirmed significantly greater overall mean ASMR-15 scores in the aASMR group [t(1,344.825) = 12.741, $p < .001$, d = .44) with a medium effect. Next, a 2 (group; aASMR vs. nASMR) x 4 (ASMR-15 subscales) MANOVA was conducted to identify any group differences in the subscales of the ASMR-15 between those who experienced authentic ASMR (aASMR) and those who experienced non-authentic ASMR (nASMR). The MANOVA was significant [λ = 0.967, F(4, 16,674) = 140.402, $p < .001$]. Analyses by subscale confirmed no group differences in the AC [F(1, 16,677) = 2.564, p = .109, η² = .000, d = 0.05] and Relax [F(1,16,677) = 3.481, p = .062, η² = .000, d = 0.05] subscales. In contrast, significantly greater scores reported by the aASMR group were observed in the Sensation [F(1,16,677) = 542.447, $p < .001$, η² = .032, d = 0.62], and the Affect subscales [F(1,16,677) = 66.370, $p < .001$, η² = .004, d = 0.23].

The ASMR-15 profile of both ASMR-sensitivity groups by was examined using paired samples t-tests by subscale. All analyses in the aASMR group were significant (all p < .019; see S1 Table). The AC score was relatively lower (mean 2.69, SD 0.93) compared to the remaining subscales confirmed with very large effect sizes (Sensation: d = 1.56; Relax; d = 2.63; Affect: d = 1.53). The relatively greater mean score in the Relax subscale (mean 4.69, SD 0.54) was a very large effect compared to both the Sensation (d = 0.98) and Affect (d = 0.90) subscales. The comparable mean scores for Sensation (mean 4.03, SD 0.78) and Affect (mean 4.05, SD 0.85) revealed a very small effect (d = 0.02). Paired samples t-tests by subscale in the nASMR group were significant (all $p < .001$). The relatively lower AC (mean 2.64, SD 0.97) was a large/

very large effect compared to the remaining subscales (Sensation: d = 0.82; Relax; d = 2.50; Affect: d = 1.24). The relatively greater mean score in the Relax subscale (mean 4.66, SD 0.60) was a very large effect compared to both Sensation (d = 1.39) and Affect (d = 1.02). The difference in mean scores between Sensation (mean 3.47, SD 1.05) and Affect (mean 3.84, SD 0.97) was a small/ medium effect (d = 0.37). See Fig 1.

To identify trait-ASMR group allocation, a k-means cluster analysis was carried out using the mean scores from the four subscales of the ASMR-15. A three-cluster grouping with 25 iterations were selected. Maximum coordinate change of .000 was achieved after 13 iterations, with participants allocated to one of three trait-ASMR groups: High (n = 7,070), Medium (n = 5,543), and Low (n = 4,066), confirmed with significant differences between cluster groups by subscale: AC [F(2, 16,676) = 11,990.914, p < .001], Sensation [F(2, 16,676) = 5,809.381, p < .001], Relax [F(2, 16,676) = 875.020, p < .001], Affect [F(2, 16,676) = 8,617.826, p < .001]. A univariate ANOVA was conducted on overall mean ASMR-15 scores by trait-ASMR group (High/ Medium/ Low). The ANOVA was significant [F(2, 16,676) = 16,042.559, p < .001, η² = .658]. Independent samples t-tests confirmed significantly greater overall mean ASMR-15 scores in the High trait-ASMR group compared to both the Medium (p < .001, d = 1.71) and Low (p < .001, d = 3.15) trait-ASMR groups. The Medium trait-ASMR group's overall mean ASMR-15 score was significantly greater than the Low trait-ASMR group (p < .001, d = 1.87). See Table 2.

Next, independent samples t-tests were conducted on the four ASMR-15 subscales by cluster group. All analyses were p < .001, except where stated (see S2 Table). The significantly greater AC score observed in the High trait-ASMR group was a very large effect compared to the Medium (d = 2.83) and Low trait-ASMR (d = 1.98) groups. In contrast, a small

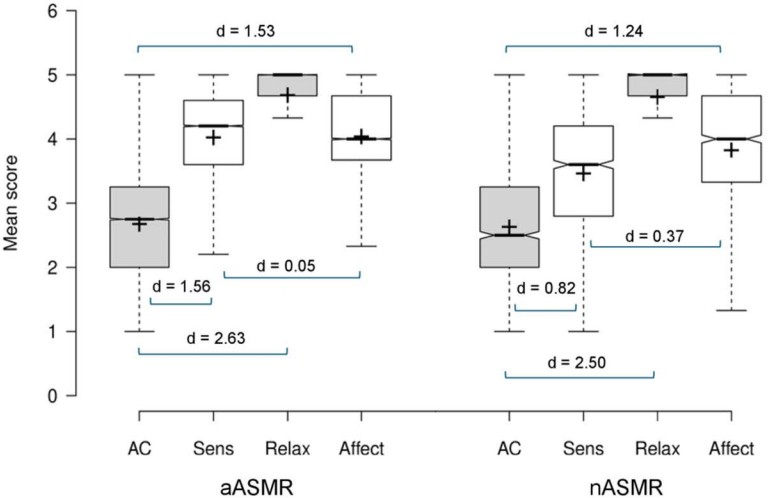

**Fig 1. ASMR-experience grouping (aASMR vs. nASMR) by ASMR-15 subscales.** Crosses indicate the mean scores, bar represents the median, whiskers extend to the 1ˢᵗ/ 3ʳᵈ interquartile range (1.5 SDs), notches represent +/-1.58 interquartile range/sqrt(n) of the mean.

**Table 2. Mean overall ASMR-15 and subscale scores by ASMR-sensitivity and trait-ASMR (SDs in parentheses).**

| Group | | n | ASMR-15 overall | Alt Consc | Sensation | Relax | Affect |
|---|---|---|---|---|---|---|---|
| ASMR-sensitivity | aASMR | 15,469 | 3.81 (0.55) | 2.69 (0.93) | 4.03 (0.78) | 4.69 (0.54) | 4.05 (0.85) |
| | nASMR | 1,120 | 3.56 (0.66) | 2.64 (0.97) | 3.47 (1.05) | 4.66 (0.60) | 3.84 (0.97) |
| Trait-ASMR | High | 7,070 | 4.23 (0.30) | 3.51 (0.54) | 4.30 (0.60) | 4.83 (0.34) | 4.50 (0.54) |
| | Medium | 5,543 | 3.74 (0.27) | 1.98 (0.53) | 4.29 (0.52) | 4.72 (0.44) | 4.21 (0.59) |
| | Low | 4,066 | 3.08 (0.42) | 2.19 (0.77) | 3.07 (0.79) | 4.41 (0.79) | 2.98 (0.72) |

effect was observed between the Medium and Low trait-ASMR groups (d = 0.32). There was no difference between the High and Medium trait-ASMR groups in the Sensation subscale (p = .193, d = 0.01), whereas both groups' scores were significantly greater compared to the Low trait-ASMR group and with very large effects (High: d = 1.75; Medium: d = 1.82). The significantly greater score in the Relax subscale observed in the High trait-ASMR group was a small effect compared to the Medium trait-ASMR group (d = 0.28) and a medium effect compared to the Low trait-ASMR group (d = 0.69). A medium effect was also observed between the Medium/ Low trait-ASMR groups (d = 0.48). Finally, for the Affect subscale, medium effect sizes were observed between the High/ Medium groups (d = 0.52), whereas very large effect sizes were observed between the High/ Low (d = 2.39), and Medium/ Low (d = 1.86) groups.

The trait-ASMR profile of each cluster was examined using paired samples t-tests by subscale and between groups (all analyses in each cluster grouping were p < .001; see S3 Table). The High trait-ASMR group's profile was driven by lower AC (mean = 3.51, SD 0.53) confirmed with very large effect sizes compared to the remaining subscales (Sensation: d = 1.34; Relax: d = 2.92; Affect: d = 1.83). The greater mean score for the Relax subscale (mean = 4.83, SD 0.34) was a very large effect compared to Sensation (d = 1.09) and a medium-to-large effect compared to Affect (d = 0.73). The difference between Sensation and Affect was a small effect (d = 0.35). The Medium trait-ASMR group's profile was driven by lower AC (mean 1.98, SD 0.53) which was a very large effect compared to the Sensation (d = 4.40), Relax (d = 5.62), and Affect (d = 3.98) subscales. Large effect sizes were observed between the Relax subscale and Sensation (d = 0.89) and Affect (d = 0.98) whereas a small effect was observed between Sensation and Affect (d = 0.14). The Low trait-ASMR group's profile was driven by lower AC (mean 2.19, SD 0.77) with very large effect sizes compared to Sensation (d = 1.13), Relax (d = 2.85), and Affect (d = 1.06). Very large effect sizes were observed between Relax and both Sensation (d = 1.57) and Affect (d = 1.89), whereas the effect size between Sensation and Affect was small (d = 0.12). See Fig 2.

In summary, scores in the Relax subscale approached a ceiling effect in all groups. When comparing between groups, the Medium and Low trait-ASMR groups' profiles were driven by significantly lower AC compared to the High group. In contrast the High and Medium groups' profiles were driven by greater Sensation compared to the Low trait-ASMR group. The Affect subscale demonstrated the greatest dimensionality, with medium effect sizes across the groups.

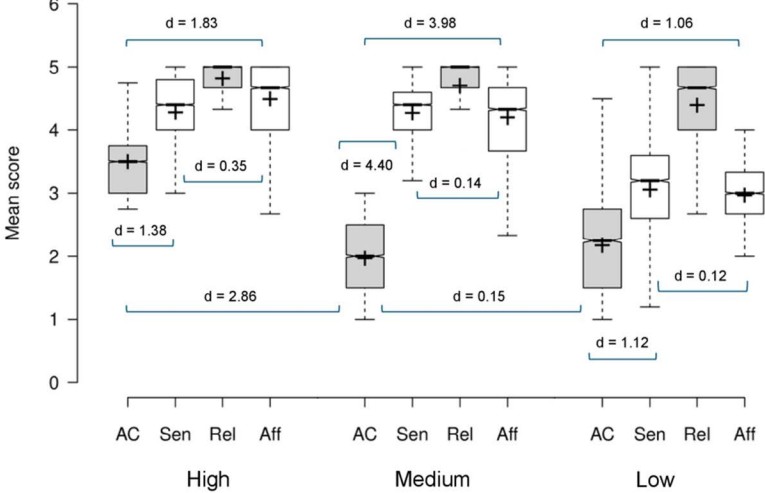

**Fig 2. Cluster grouping for the trait-ASMR groups by ASMR-15 subscale.** Crosses indicate the mean scores, bar represents the median, whiskers extend to the 1st/ 3rd interquartile range (1.5 SDs), notches represent +/-1.58 interquartile range/sqrt(n) of the mean.

## Interim summary

The pattern of results indicate that cluster differentiation based on the four subscales of the ASMR-15 is much more informative than a self-classification process. These results provide strong evidence that a data-driven dimensional approach to classifying trait-ASMR provides richer information about this individual difference.

## Trigger endorsement and intensity ratings as a function of ASMR categorisation

Thus far, we have demonstrated how ASMR propensity differs depending on individual differences in ASMR characteristics. Next, we investigated trigger endorsement and ASMR intensity depending on trait-ASMR and ASMR-sensitivity.

Group differences in trait-ASMR and ASMR-sensitivity in the trigger endorsement and intensity ratings of 58 different triggers was explored via four univariate ANOVAs, with trait-ASMR and ASMR-sensitivity as the grouping variables, and total trigger endorsement and mean trigger intensity ratings as the DVs. Analyses with the total number of triggers endorsed as the DV revealed a main effect of trait-ASMR group [$F_{(2, 16,676)} = 405.337$, $p < .001$, $\eta^2 = .046$]. All group differences were $p < .001$, and small to medium effect sizes (High/ Medium $d = 0.24$; High/ Low $d = 0.56$; Medium/ Low $d = 0.33$). The ASMR-sensitivity group difference was also significant [$F_{(1, 16,677)} = 26.838$, $p < .001$, $\eta^2 = .002$] but a small effect ($d = 0.16$). Analyses with the overall mean trigger intensity rating score as the DV also found a main effect of trait-ASMR group [$F_{(2, 16,676)} = 579.069$, $p < .001$, $\eta^2 = .065$]. All group differences were $p < .001$ and supported with small to medium effect sizes (High/ Medium $d = 0.32$: High/ Low $d = 0.66$, Medium/ Low $d = 0.35$). The main effect of ASMR-sensitivity on trigger intensity rating was also significant [$F_{(1, 16,677)} = 20.667$, $p < .001$, $\eta^2 = .001$] and also a small effect ($d = 0.15$). See Table 3.

## Principal component analysis

Finally, we have taken a data-driven approach using PCA to categorise triggers and explore any differences in categorisation based on ASMR-propensity. PCA using Varimax rotation was conducted to classify the 58 triggers into meaningful dimensions using the trigger intensity rating scores. Criteria were set at ≥.400 for inclusion, and any item loading onto more than one factor at ≥.300 was discounted [38]. The initial PCA revealed 13 factors and with considerable cross-loading (see S4 Table). Six iterations were required to meet the criteria, with the final iteration identifying 8 factors with eigenvalues of ≥ 1.0, across 25 remaining triggers, and which accounted for 57.96% of the variance. The items loading onto these factors aligned with the trigger grouping labels previously identified [13,14] (see Table 4). The point of inflexion on the curve of the scree plot indicated four factors of importance which contributed to 39.36% of the variance: *Roleplay*, 17.74%; *Watching*, 9.00%; *Interpersonal care (IPC)*, 7.02%, and *Visual*, 5.60%.

Inspection of the 33 items removed from PCA01, due to cross-loading of ≥.300, identified multiple triggers that are frequently endorsed in both the literature and within the ASMR community forums as eliciting ASMR. A second PCA was run on these 33 items with inclusion criteria of ≥.400. Two iterations were required as trigger 44 *('Rude/ demeaning to the viewer')* cross loaded with 5 categories but did not meet criteria. No further exclusion criteria were applied to reveal the extent of any cross-loading of the remaining 32 triggers (see Table 5). Any cross-loading of ≤.200 was considered a weak

**Table 3. Mean triggers endorsed and trigger intensity rating by ASMR-sensitivity and trait-ASMR (SDs in parentheses).**

| | Group | n | Mean no. triggers endorsed | Mean trigger intensity |
|---|---|---|---|---|
| ASMR-sensitivity | aASMR | 15469 | 24.43 (10.16) | 0.94 (0.41) |
| | nASMR | 1210 | 22.86 (10.07) | 0.88 (0.40) |
| Trait-ASMR | High | 7070 | 26.47 (10.39) | 1.04 (0.43) |
| | Medium | 5543 | 24.04 (9.69) | 0.91 (0.38) |
| | Low | 4066 | 20.93 (9.39) | 0.78 (0.36) |

**Table 4. PCA01 with factor loadings, cross loadings, and percentage of variance of 25 triggers.**

| | Roleplay 17.74% | Watching 9.00% | IPC 7.02% | Visual 5.60% | Tactile 5.16% | Non-voc auditory 4.71% | Vocal auditory 4.53% | Other 4.19% |
|---|---|---|---|---|---|---|---|---|
| Interviewing for a job | .749 | | | | | | | |
| Taking a survey | .712 | | .138 | | | | | .103 |
| Q&A roleplay | .701 | | | | | | .123 | .104 |
| Sales roleplay | .663 | .253 | .125 | | | | | |
| Getting assistance from teacher | .550 | .219 | .195 | | | | .181 | |
| Watching sort through items | .115 | .769 | | | | .116 | | |
| Watching unwrap something | | .703 | | | | .214 | | |
| Watching perform an activity | .124 | .682 | | | .111 | | | .122 |
| Towel folding tutorials | .212 | .562 | | .216 | .116 | −.134 | | −.106 |
| Watching read/ browse | .193 | .545 | | | | | .123 | .218 |
| Cranial nerve exam | .194 | | .867 | | | | | |
| Eye exam | .226 | | .863 | | | | | |
| Gentle hand movements | | | | .861 | | .103 | | .105 |
| Slow movements | | .136 | | .844 | | | | |
| Hair or scalp touching | | | | | .864 | .131 | | |
| Scalp or back massage | | | | | .864 | .102 | | |
| Scratching sounds | | | | | .228 | .729 | | |
| Tapping sounds | | | | .224 | | .728 | | |
| Crisp sounds | | .237 | .143 | | | .558 | .115 | |
| Speaking in a foreign language | | | | | | | .774 | .169 |
| Speaking in an accent | .147 | .145 | .104 | | | | .762 | |
| Rambling | .217 | | −.130 | | | .107 | .460 | .146 |
| Colored lights changing color | | | .169 | .204 | | | | .657 |
| Nature sounds | | | | | .176 | | | .628 |
| Movie/software sound effects | .154 | .110 | | | | | | .607 |

relationship. The second PCA also identified 8 factors with eigenvalues of ≥1.0 and which accounted for 52.70% of the variance. The items loading onto the first six factors aligned with the trigger grouping labels identified in the PCA01. The point of inflexion on the curve of the scree plot indicated four factors of importance and which contributed to 37.73% of the variance: *Roleplay,* 18.86%; *Interpersonal care (IPC),* 7.49%; *Tactile*, 6.37%; and *Watching* 5.01%.

The factor loadings were saved as Anderson-Rubin standardised scores (see S5 Table). To identify whether factor loading scores in both PCAs could be explained by trait-ASMR and/ or ASMR-sensitivity categorisation, four MANOVAs were run with trait-ASMR (High/ Medium/ Low) and ASMR-sensitivity (aASMR/ nASMR) as the grouping variables and the Anderson-Rubin standardised scores for each category across PCA01 and PCA02 as DVs (see S6 Table).

The first MANOVA based on trait-ASMR grouping and the factor loading scores from PCA01 (DVs) was significant [λ = .930, F(16, 33,338) = 76.742, p < .001, η² = .036] with significant trait-ASMR group effects observed in all trigger categories (all p < .001, η² ≤ .014). Pairwise comparisons with Bonferroni corrections (p = .05/3) confirmed significantly greater factor loadings in all trigger categories in the High trait-ASMR group compared to both the Medium and Low groups (all p ≤ .006, d ≤ 0.30) except for the High/ Low group difference in the *Other* category (p = .052, d = 0.11). Significantly greater factor loadings were observed in the Medium compared to the Low trait-ASMR group in all categories (all p < .001, d ≤ 0.21).

**Table 5. PCA02 with factor loadings, cross loadings, and percentage of variance for the remaining 32 triggers.**

| | Roleplay 18.86% | IPC 7.49% | Tactile 6.37% | Watching 5.01% | Non-voc auditory 4.64% | Other auditory 3.72% | Other 3.44% | Vocal auditory 3.17% |
|---|---|---|---|---|---|---|---|---|
| Clinical roleplay | .792 | .155 | | | | | | |
| Medical appointments | .741 | | | | .125 | | | |
| Consultation roleplay | .729 | .125 | | | | | | .146 |
| Pharmacist consultation | .704 | | | .195 | | | | |
| Customer service roleplay | .684 | | | .150 | −.138 | | .103 | .197 |
| Roleplay – Inter-personal attention | .504 | .287 | .180 | | .198 | | | .118 |
| Hairstyling/ braiding | .146 | .738 | | | | .104 | | |
| Face product application | .231 | .678 | .310 | | | | .130 | |
| Washing/ cutting hair | .152 | .658 | | | | .238 | | |
| Beauty care roleplay | .474 | .560 | .179 | | | | .115 | .109 |
| Makeup brushes on face/body | | .541 | .514 | | .108 | | | |
| Face touching or tapping | .114 | .318 | .664 | | .238 | | | |
| Tracing skin with finger | | .236 | .656 | | | | | |
| Someone swaying | | −.155 | .531 | .217 | | | .132 | .112 |
| Hand care | .176 | .298 | .461 | .164 | −.110 | | .217 | .118 |
| Watching play a game | | | | .691 | | | | |
| Device/ video game demo | .130 | | | .586 | | | .112 | .101 |
| Watching paint/ draw | | .301 | | .578 | | .244 | | .155 |
| Magic tricks | | | .139 | .529 | | | | −.131 |
| Being sketched/ painted | .243 | .386 | .129 | .398 | .139 | .133 | −.148 | |
| Inaudible whispering | | | | | .709 | | .156 | |
| Binaural sounds | | | .163 | | .653 | .173 | | |
| Repetitive sounds | | | .386 | | .392 | .285 | .109 | |
| Item sounds: pencils/beads/rice | | .146 | .201 | .138 | | .698 | .134 | |
| Crinkling sounds | | | | | | .612 | .383 | |
| Mechanical sounds | .265 | .190 | −.162 | .254 | | .554 | | |
| Sticky finger sounds | | | .135 | | | .188 | .701 | |
| Squishing sounds | | | | | | .197 | .656 | |
| Mouth sounds | | | −.132 | .159 | .392 | −.303 | .548 | |
| Soft speaking | .228 | | .104 | | −.177 | | | .670 |
| Audible whispering | | | | | .370 | | | .658 |
| Reading | | | | .373 | | | | .587 |

The second MANOVA based on ASMR-sensitivity and factor loading scores from PCA01 (DVs) was significant [$\lambda = .994$, $F(8, 16,670) = 12.193$, $p < .001$, $\eta^2 = .006$]. Analyses by trigger category were significant for *Watching, IPC, Tactile*, and *Other* ($p \le .004$, $d \le 0.16$). Non-significant group differences were observed for *Roleplay, Visual, Non-vocal auditory,* and *Vocal-auditory* ($p \ge .129$, $d \le 0.05$). See Fig 3.

The third MANOVA based on trait-ASMR grouping and the factor loading scores from PCA02 (DVs) revealed a significant main effect [$\lambda = .936$, $F(16, 33,338) = 70.076$, $p < .001$, $\eta^2 = .033$] with significant group effects observed in all trigger categories (all $p < .001$, $\eta^2 \le .014$). Pairwise comparisons with Bonferroni corrections ($p = .05/3$), confirmed significantly greater factor loadings in the High trait-ASMR group compared to the Medium group in all trigger categories ($p \le .006$, $d \le 0.15$) except for *Non-vocal auditory* ($p = .409$, $d = 0.03$). The factor loadings in the High group were significantly greater than the Low group across all factors (all $p < .001$, $d \le 0.29$). Factor loadings were significantly greater

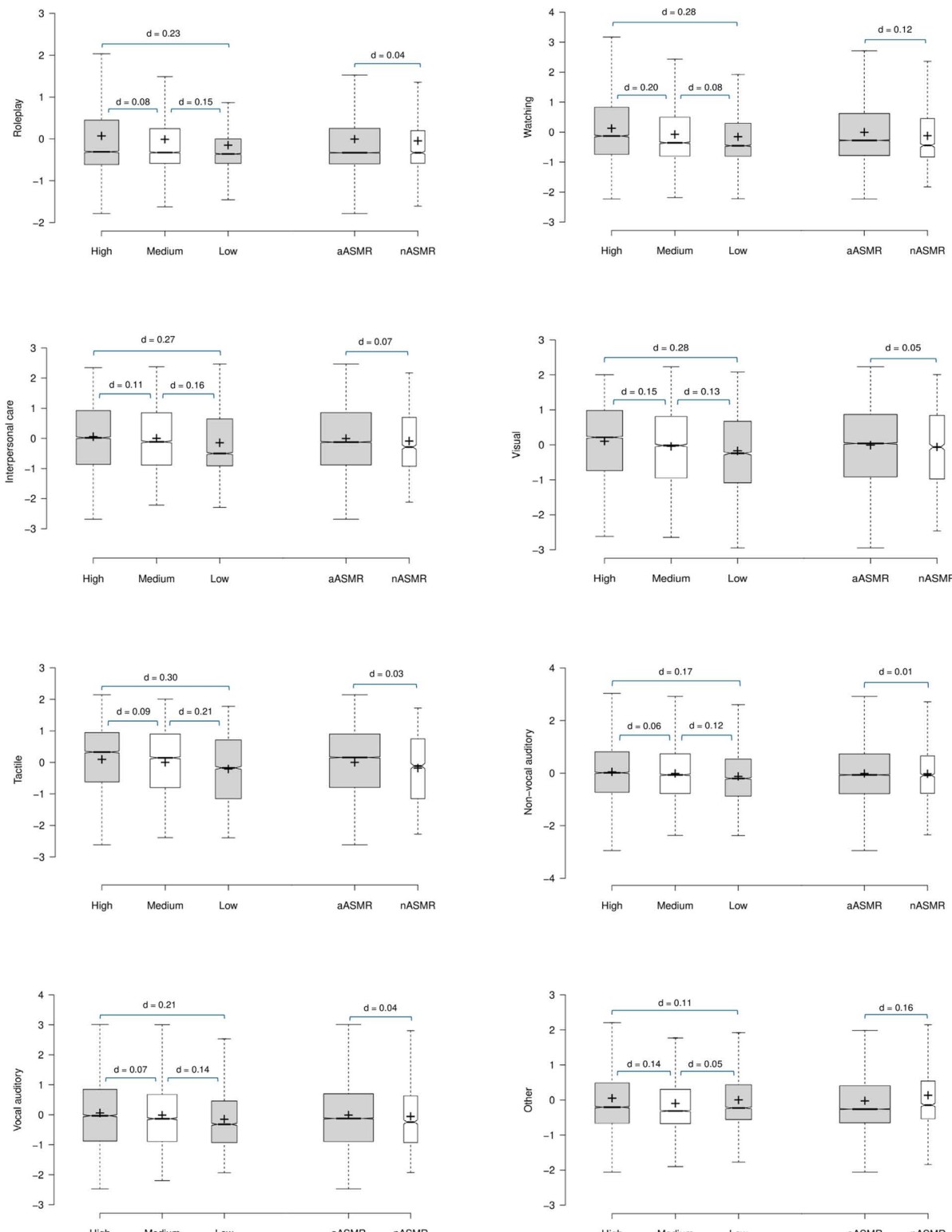

**Fig 3. PCA01: Mean factor loadings for each trigger category by trait-ASMR and ASMR-sensitivity.** Crosses indicate the mean scores, bar represents the median, whiskers extend to the 1st/ 3rd interquartile range (1.5 SDs), notches represent +/-1.58 interquartile range/sqrt(n) of the mean.

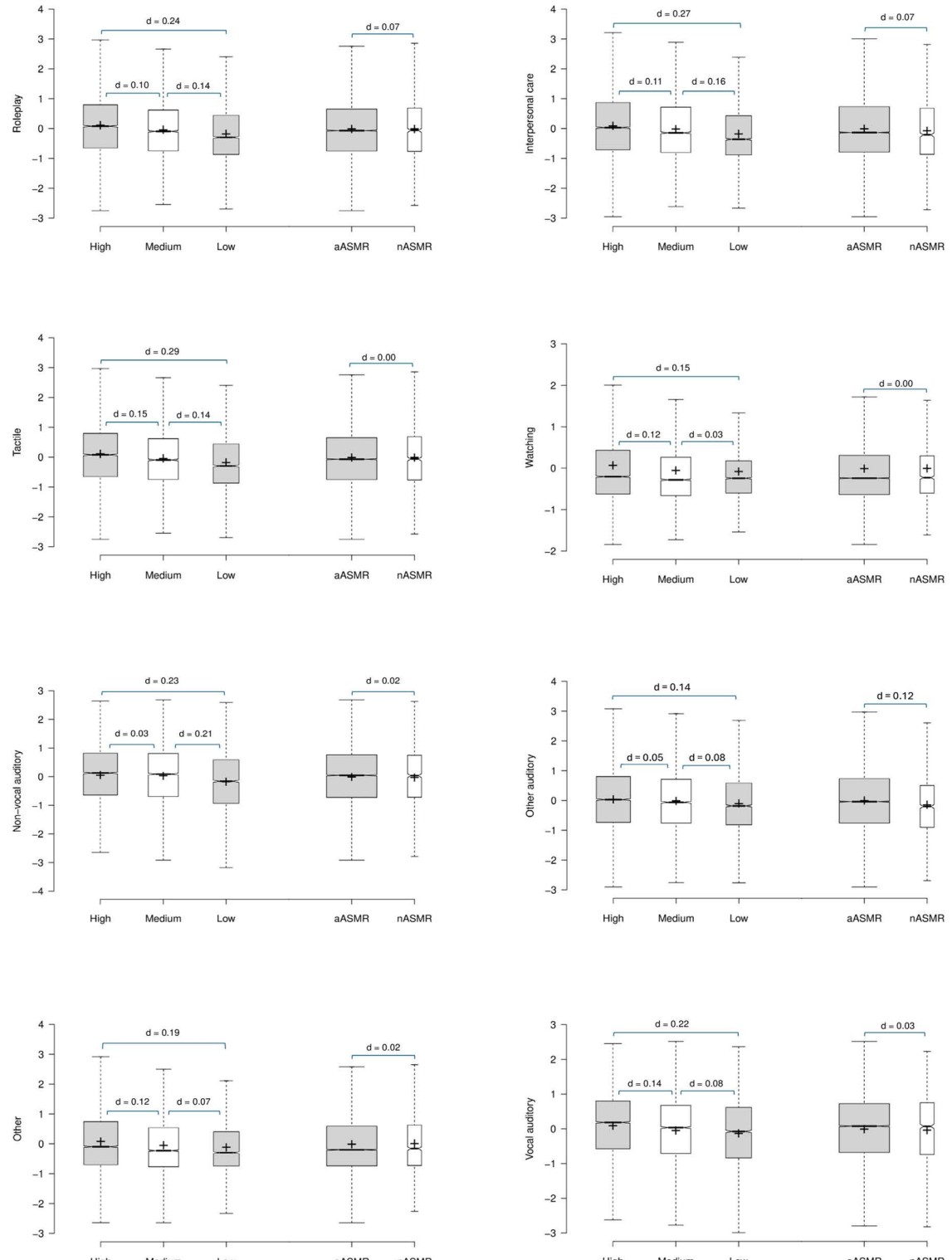

**Fig 4. PCA02: Mean factor loadings for each trigger category by trait-ASMR and ASMR-sensitivity.** Crosses indicate the mean scores, bar represents the median, whiskers extend to the 1st/ 3rd interquartile range (1.5 SDs), notches represent +/-1.58 interquartile range/sqrt(n) of the mean.

in the Medium compared to the Low group in all trigger categories (p ≤ .005, d ≤ 0.21) except for *Watching* (p = .704, d = 0.03).

The fourth MANOVA based on ASMR-sensitivity with loading scores for PCA02 as the DVs was significant [λ = .998, $F(8, 16,670) = 4.346$, p < .001, η² = .002]. Analyses by trigger category revealed significant group effects for *Roleplay, IPC*, and *Other Auditory* (p ≤ .023, d ≤ 0.12). Non-significant group differences were observed for *Tactile, Watching, Non-vocal auditory, Other,* and *Vocal-auditory* (p ≥ .343, d ≤ 0.07). See Fig 4.

In summary, two PCAs on trigger intensity ratings identified four key categories of triggers: *Roleplay, Interpersonal care, Tactile*, and *Watching*. Analyses of the Anderson-Rubin standardised scores revealed much stronger group differences in trigger categorisation based on trait-AMSR compared with ASMR-sensitivity.

## Discussion

The aim of the current study was to: a) examine the profile of trait-ASMR within the ASMR-viewing community by using the psychometric properties of the ASMR-15 [8] (High/ Medium/ Low trait-ASMR) compared to the conventional process of self-reported ASMR-sensitivity (authentic – where tingles originate in the head and/ or neck vs. non-authentic – where tingles originated elsewhere in the body); b) investigate the classification of ASMR triggers from a data-driven perspective using PCA; and c) explore whether differences existed in trigger categorisation dependent on trait-ASMR and ASMR-sensitivity categorisations. Due to the size of the sample, interpretation of the findings was made predominantly based on effect sizes, and significant/ non-significant differences are only emphasised where meaningful.

The first aim of the study was to address the relationship between a conventional self-classification process (authentic location versus non-authentic) and a cluster analysis classification derived from the scores on the ASMR-15. The two classification approaches do not lead to similar profiles of ASMR-15 scoring. Firstly, as expected, most participants (~93%) reported experiencing 'authentic' ASMR where tingles originated in the head and/ or neck [1] however, as shown in Fig 1, what is striking about the aASMR and nASMR groups' ASMR-15 profiles is their similarity. When group categorisation is based solely on location of tingles, our findings indicate that ASMR-viewers do not differ across altered perceptual states (AC) or the psychological response (Relax), and report minimal differences in the socio-emotional aspects of the experience (Affect). Group differences were only driven by significantly greater scores in the Sensation subscale in the aASMR group, accompanied with a medium effect. This subscale relates specifically to a physiological response when experiencing ASMR. Considering ASMR-sensitivity group allocation was based on participants' self-reported location of tingles, this finding is not surprising, especially as two of the items in this subscale refer specifically to sensations in the head [8].

In contrast, a cluster analysis approach on the ASMR-15 scores more readily identifies differential profiles across the three (High/ Medium/ Low) cluster groups. First, considering relaxation is a focal reason many ASMR-viewers seek out ASMR [1], the high scores in the Relax subscale in all groups was not unexpected. The Low group was characterised by relatively depressed Sensation and Affect scores in addition to a low AC score. The relatively higher AC score in the High group differentiates this group from the Medium group, with a very large effect size found and is consistent with previous research [8,11]. The High and Medium groups' profiles were equally driven by the physiological aspects of the experience (Sensation) evidenced by a small effect, whereas the emotional aspect (Affect) presents greater dimensionality within the ASMR-viewing community, with medium effects between all groups. Notably, this contradicts qualitative perspectives from ASMR-users who stated the psychological impact was of greater importance over the somatic experience of ASMR [28].

When looking at the breakdown between the ASMR-sensitivity and the trait-ASMR groups, an informative pattern emerged (see S7 Table). Despite accounting for much of the overall sample, only 43% of participants in the aASMR group were High in trait-ASMR, while nearly a quarter of this group (23.1%) were Low in trait-ASMR. In contrast, the nASMR group accounted for only 7% of the sample, however over one third (34.5%) of this group were also High in trait-ASMR and 41% were Low in trait-ASMR. This further supports the limited literature to date which has evidenced that a

dimensional approach based on trait-ASMR characteristics is more informative than a categorical approach based solely on self-reporting of ASMR experience [17,33].

When participants were asked to endorse triggers that elicited ASMR and rate their tingle intensity, distinct patterns emerged. The ASMR-sensitivity (aASMR vs nASMR) group differences in the number of triggers endorsed and trigger intensity ratings were both accompanied by small effect sizes (cf. [2]). Considering the size of the sample, these findings are notable. In contrast, the increasing number of triggers endorsed and tingle intensity observed with increasing levels of trait-ASMR was supported with an overall medium effect size, consistent with previous research [14]. Indeed, these findings demonstrate that, within the ASMR-viewing community, the physiological response that defines the phenomenon [1] is not of such importance regarding trigger endorsement or intensity. Rather, individual levels of ASMR-responsivity are informative, demonstrating that the physiological experience of ASMR is not ubiquitous across this community.

The second aim was to identify, through PCA, the nature of the trigger characteristics experienced by the ASMR viewers. Across the two PCAs, four key categories emerged: *Roleplay, Watching, Interpersonal care,* and *Tactile.* These categorisations partially support previous attempts to classify triggers [13,14], however we have demonstrated that *Roleplay* is a key category, and that previously combined classifications (*Tactile/ IPC*) are in fact discrete. Furthermore, it concurs with fMRI neuroimaging which has observed distinct brain activity across categories that align with those in the current study [39].

*Roleplay* emerged as the most dominant category in both PCAs despite previous analysis of ASMR content which found ~72% do not include any form of roleplay [22]. Poerio et al. [14] excluded *Roleplay* in their categorisation due to its specificity with ASMR online communities, and the results here clearly support this association – though we demonstrate the importance of its inclusion as a trigger category. Many triggers in the current study only cross-loaded weakly; only one trigger, *Beauty care roleplay*, while loading onto the *IPC* category in PCA01, had an almost equal loading onto the *Roleplay* category. Additionally, our results identified distinct differences in the themes between the two analyses across categories. The triggers loading onto the *Roleplay* category in PCA01 appeared to have situation-specific context. Three of the triggers loading onto *Roleplay* in PCA02 have a clinical context, however the triggers loading onto *IPC* in PCA01 were also clinical in nature (e.g., *cranial nerve exam*). *Roleplay* emerging as a distinct category raises interesting questions. While non-roleplay videos present an ASMRtist performing routine triggering activities (e.g., tactile gestures, microphone brushing, manipulating objects), roleplay videos present a simulated interaction. These engage the viewer from a first-person perspective with face-to-face personal attention and include a typically whispering dialogue that implies the viewer is present and participating in an intimate conversation with the ASMRtist [18]. This is further emphasised by analysis of viewer comments which has found *Roleplay* videos garner greater intimate commentary [22]. A key category across both PCAs, *Watching,* also concurs with Niu et al. [22] who identified sensory-rich observational activities as one of the three central features of ASMR content. A previous study found *Watching* to be the second most frequently endorsed category (~43%), though there was only one trigger which covered this category more generally [11]. Fredborg et al. [13] also found triggers that involved the participant as a third-party viewer loaded strongly onto their *Watching* category and were distinct from triggers that involved watching activities that were either tactile or involved some form of personal attention.

*Interpersonal care* and *Tactile* categories were identified as discrete categories (cf. [14]) though with an element of cross-loading in PCA02. The triggers loading onto *IPC* tended to have a beauty element (*Hairstyling/ Makeup application*) whereas the triggers loading onto the *Tactile* category predominantly reflected scenarios related to contact with the face. The weakest loading trigger in the *Tactile* category (*Hand Care*) cross-loaded weakly across most of the other factors. These categories partly align with Roberts et al. [11] who found ~50% of participants endorsed triggers associated with close personal attention, and Poerio et al. [14] who also reported the greatest intensity ratings to triggers that included physical contact with the body. In the current study, the trigger that specifically referred to contact with the body (*Scalp/ Back massage*) was endorsed by 66.6% of the sample also loaded strongly onto the *Tactile* category in PCA01. The emergence of *Tactile* as a key trigger category further supports the suggestion that the visual-auditory element of ASMR

videos may act as cues for the affective responses associated with AT [16]. The associated slow-conducting unmyelinated tactile fibres (CT fibres) project to subcortical regions associated with the encoding of bodily signals [40,41]. This aligns closely with affective haptics which investigate the 'computational aspects of mediated affective touch' [42] (pp. 27), with haptic audio-visuality posited as a manner of touching with the eyes and ears [43]. Evidence for this theoretical argument was provided in a recent study incorporating a virtual reality (VR) paradigm [44]. Participants engaged in a two-person VR interaction with visuo-auditory ASMR triggers as well as vibrotactile feedback. The authors observed improved positive affective responses including relaxation, calmness, soothing, and enjoyment, with some ASMR-naïve participants experiencing a tingling sensation indicative of authentic ASMR in response to the auditory element. Furthermore, some participants also reported a vibration feedback sensation in the head despite the feedback devices only being worn on hands, arms, and the body. Similarly, Gillmeister et al. [17] found increased trait-ASMR was positively associated with affective reactions to social touch, as well as more recurrent and intense tactile sensations in response to viewing content that portrayed human touch.

Inspection of the categories that appeared to be less informative in our findings raise interesting questions about a data-driven analysis vs. self-reported trigger preference. *Non-vocal auditory*, again aligned with triggers identified by Fredborg et al. [13] including *Tapping* and *Scratching*. However, *Non-vocal auditory* only accounted for 4.71/ 4.64% of the variance respectively in our analyses. Similarly, the *Vocal auditory* category was weak in both PCAs despite the triggers loading strongly onto this factor. Of note, 'Audible whispering' also cross-loaded strongly with the *Non-vocal auditory* category. Roberts et al. [11] found *Whispering* and *Soft speaking* to be the third and fourth most endorsed triggers, while 75% participants in the seminal paper by Barratt and Davis [1] reported to be triggered by *Whispering*. The act of whispering infers closeness and intimacy [45] and ASMR is considered a form of digital intimacy [46]. However, participant feedback indicates that contextual factors need to be considered regarding this trigger [8]. Participants responded more positively when they intentionally engaged with AMSR videos that included whispering compared with those that included inadvertent whispering – the latter often evoking negative affect similar to misophonia [47].

Following the identification of trigger categories, the third aim was to identify putative associations between the classification groups and the trigger characteristics as revealed by the PCA. The MANOVA based upon ASMR-sensitivity revealed trivial associations, with $\eta^2$ effect sizes all well below .01. In contrast the MANOVA with the cluster groups revealed an effect size of a magnitude 10 times greater than that associated with the ASMR-sensitivity groups. When comparing PCA factor loading by ASMR-sensitivity and trait-ASMR groups, interesting patterns emerged. First, the differences in factor loading scores between the aASMR and nASMR groups were typically very small and even negligible effect sizes except for *IPC* in PCA01. This again demonstrates that, within this community, experiencing 'authentic' ASMR is not of key importance [5]. When comparing trait-ASMR group differences, the *Tactile* factor in both analyses demonstrated the greatest dimensionality with small but equivalent effect sizes across the trait-ASMR groups. The *Role-play* category was also characterised by an element of dimensionality about both situation-specific and clinically focused triggers (cf. [33]). In contrast, non-significant differences were observed between the Medium/ Low groups in the *Watching* category in PCA01, and the effect size between these groups was extremely small in both analyses. This indicates distinct differences depending on the viewer's active or passive engagement, where those higher in trait-ASMR have higher factor loadings where the viewer is the recipient within the scenario (first-person perspective), and those medium and lower in trait-ASMR more likely to endorse a third-person perspective. Similarly, distinct patterns emerged between the two analyses regarding *Vocal-auditory* category. PCA01 included less frequently endorsed triggers, and where an extremely small effect size was observed between the High/ Medium groups. In contrast, PCA02 included the more commonly endorsed triggers such as *Soft speaking* and *Audible whispering*, and a negligible effect size was observed between the Medium/ Low groups [4,13].

Despite the unexpected strength of our sample size, there were some notable limitations to our study. Given that we specifically targeted current ASMR-viewers, our findings are only relevant to this population, but did not account for prior ASMR

exposure or other potential confounds in individual differences such as personality traits, anxiety, or sensory processing sensitivity [31,48]. There was a potential bias in our respondents' trigger endorsement and intensity ratings as most of our participants were directed to our survey via a link posted by a single ASMR channel on YouTube. Whilst this channel's audience might reflect a large proportion of the ASMR community, it is likely not representative of all ASMR experiencers. As such, the trigger categories identified in our PCAs and trait-ASMR profiles in the factor loadings may reflect selective bias towards the style of ASMR content created by this ASMRtist. To address these issues, future studies should prioritize representativeness during sampling in order to provide a broader perspective of trait-ASMR across both the ASMR-viewing and ASMR-naïve populations. The ASMR-15 [8] is currently the only available psychometric measure of trait-ASMR, however the ceiling effect observed in the Relax subscale here and in other research [8,49] indicates biased responding when used with the ASMR-viewing population and raises questions as to how useful this subscale is particularly within a targeted sample such as those recruited in the current study. Furthermore, we restricted our cluster analysis from the ASMR-15 to three trait-ASMR categories for ease of interpretation, and due to the source of recruitment for the sample, the effect sizes between groups were generally small. Also, participants were allocated to the aASMR/ nASMR group based on self-reported tingle location. Whilst we have argued against using tingle location to delineate ASMR experience based on the consistent overall very small effect sizes between our aASMR and nASMR groups, a data-driven approach to categorise ASMR-sensitivity (see [2]) may have identified nuances we did not observe here. A final key limitation relates to the descriptions of the triggers which itself was a form of categorisation pre-analysis (e.g., 'Sales *Roleplay*'/ '*Watching* someone sort through items'). Also, many commonly endorsed triggers loaded onto categories that only accounted for very small variance in the PCAs, indicative of the subjective nature of individual ASMR trigger preferences. Greater research incorporating objective measures, such as arousal identified via GSR, physiological, and neuroimaging techniques (e.g., [50–52]), is required to substantiate these individual differences in both ASMR-propensity and trigger endorsement.

## Conclusion

In conclusion, the findings from the current study demonstrate that whispering minds do not always tingle alike. Our study adds to the ASMR literature by expanding our current understanding of individual differences related to experiencing ASMR. We have identified meaningful categories within in a targeted population of ASMR-viewers who access ASMR content on a regular basis. This study has provided the first opportunity to profile, from a data-driven perspective, characteristics of the ASMR phenomenon in an extensive sample of this niche population. We acknowledge that the categories generated from the PCAs are restricted to the stimuli used in the current study. However, the findings have provided the first opportunity to unpick nuances of trait-ASMR and trigger classification within a very large sample of the ASMR-viewing population. We hope the results here will prompt even more data-driven approaches to investigating this phenomenon since interest in ASMR does not seem to be fading.

## Supporting information

**S1 Table. Paired samples t-tests by ASMR-15 subscales and by ASMR-sensitivity groups.**
(DOCX)

**S2 Table. Independent samples t-tests by subscales of the ASMR-15 between cluster groups.**
(DOCX)

**S3 Table. Paired samples t-tests by ASMR-15 subscales between trait-ASMR cluster groups.**
(DOCX)

**S4 Table. Full Principal Component Analysis.**
(DOCX)

**S5 Table. Factor loadings and effect sizes for PCA01 and PCA02 by trait-ASMR and ASMR-sensitivity groups.**
(DOCX)

**S6 Table. Inferential statistics for two principal component analyses by trait-ASMR and ASMR-sensitivity groups.**
(DOCX)

**S7 Table. Number and percentage of participant group allocation by ASMR-sensitivity and trait-ASMR.**
(DOCX)

## Author contributions

**Conceptualization:** Joanna M. H. Greer, Colin J. Hamilton.

**Data curation:** Daniela Beckelhymer, Emily Thompson.

**Formal analysis:** Joanna M. H. Greer.

**Investigation:** Daniela Beckelhymer, Carin Perilloux.

**Methodology:** Joanna M. H. Greer.

**Project administration:** Daniela Beckelhymer, Emily Thompson.

**Supervision:** Carin Perilloux.

**Visualization:** Joanna M. H. Greer, Colin J. Hamilton.

**Writing – original draft:** Joanna M. H. Greer.

**Writing – review & editing:** Joanna M. H. Greer, Colin J. Hamilton, Daniela Beckelhymer, Emily Thompson, Carin Perilloux.

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
