## [Decision Letter · Decision Letter 0]

PONE-D-25-05327Do whispering minds tingle alike? Exploring the relationship between ASMR-sensitivity, trait-ASMR, and trigger preferencePLOS ONE

Dear Dr. Greer,

Thank you for submitting your manuscript to PLOS ONE. After careful consideration, we feel that it has merit but does not fully meet PLOS ONE’s publication criteria as it currently stands. Therefore, we invite you to submit a revised version of the manuscript that addresses the points raised during the review process.

We look forward to receiving your revised manuscript.

Kind regards,

PUGAZHENTHAN THANGARAJU, M.D.,Ph.D., FRCP (LONDON)., FRCP (GLASGOW).,MBA.,

Academic Editor

PLOS ONE

4. We note you have included a table to which you do not refer in the text of your manuscript. Please ensure that you refer to Table 1 and 2 in your text; if accepted, production will need this reference to link the reader to the Table.

Additional Editor Comments (if provided):

Reviewers' comments:

Reviewer's Responses to Questions

**Comments to the Author**

1. Is the manuscript technically sound, and do the data support the conclusions?

Reviewer #1: Yes

Reviewer #2: Partly

Reviewer #3: Partly

2. Has the statistical analysis been performed appropriately and rigorously? 

Reviewer #1: Yes

Reviewer #2: Yes

Reviewer #3: Yes

3. Have the authors made all data underlying the findings in their manuscript fully available?

Reviewer #1: Yes

Reviewer #2: No

Reviewer #3: Yes

4. Is the manuscript presented in an intelligible fashion and written in standard English?

Reviewer #1: Yes

Reviewer #2: No

Reviewer #3: No

5. Review Comments to the Author

Reviewer #1: 1. Consider providing a clearer theoretical rationale as to why trait-ASMR should be prioritized over ASMR-sensitivity in explaining individual differences.

2. The identification of four trigger categories through PCA is a valuable contribution, but how do these findings align with prior trigger classification frameworks? A more detailed comparison would be beneficial.

Reviewer #2: Dear Authors,

This research employs a data-driven approach using PCA, k-means clustering, and MANOVA, which adds valuable insights to the ASMR field. However, several aspects require further clarification and improvement.

Please consider the following modifications (Major Revisions):

1. Justification of Statistical Approaches

-While k-means clustering, PCA, and MANOVA are well-suited for your study, please provide a justification for using these methods over alternative clustering or dimensionality reduction techniques such as hierarchical clustering or factor analysis.

-Additionally, discuss the assumptions and limitations of PCA in categorizing ASMR triggers, particularly how it handles cross-loading variables and whether alternative factor rotation methods (e.g., oblique rotation) were considered.

2. Handling of Cross-Loading in PCA

-The PCA analyses identified significant cross-loadings, leading to the exclusion of multiple triggers.

-Please clarify why a higher threshold (≥0.400) for factor inclusion was chosen and whether alternative methods for handling cross-loadings (e.g., exploratory factor analysis with confirmatory validation) were considered.

3. Effect Size Interpretation and Practical Significance

-Given the large sample size (n = 16,679), almost all comparisons were statistically significant. However, the effect sizes (Cohen’s d, η²) are often small to medium, making practical significance unclear.

-Please emphasize which findings are most meaningful in real-world ASMR categorization rather than focusing solely on statistical significance.

4. Potential Confounding Variables

-The study does not account for external factors that may influence ASMR perception, such as prior exposure to ASMR, neuroticism, anxiety levels, or sensory processing sensitivity.

-If these variables were not measured, acknowledge them as limitations.

Trait-ASMR Classification and Cluster Validity

-The k-means clustering approach categorized participants into High, Medium, and Low trait-ASMR groups.

Please clarify whether alternative clustering solutions (e.g., a four-cluster or two-cluster model) were tested and how the final three-cluster solution was determined as optimal.

5. Demographic Influence on ASMR Experience

-The study includes a diverse sample but does not analyze whether age, gender, or ethnicity influenced ASMR sensitivity or trait-ASMR classification.

-Consider running additional subgroup analyses to determine if demographics significantly impact ASMR experience and trigger preference.

6. Clarification of "Authentic" vs. "Non-Authentic" ASMR

-The distinction between "Authentic ASMR" (tingles originating in the head/neck) and "Non-Authentic ASMR" (tingles elsewhere) is interesting but may not fully capture the subjective variability of ASMR experiences.

-Please discuss whether alternative ASMR classifications (e.g., intensity-based or multidimensional grouping) were considered.

7.Consider discussing any psychometric limitations (e.g., response bias, scale ceiling effects).

8. Discussion of Ethical Considerations

Since the study recruited participants through social media platforms, please briefly mention any ethical considerations (e.g., potential biases in self-selection, data privacy concerns).

Reviewer #3: Comments/queries

1. Kindly make a structured abstract

2. In all the figures both mean and median scores are evaluated and incorporated in the same image. Explain why it is done.

3. Expand the abbreviations wherever it is used initially

4. Kindly attach the ASMR-15 tool in the appendix.

5. If permission is obtained to use ASMR-15 tool, kindly mention it

6. The Methodology section of the manuscript require reorganization of contents. Draft it in a sequential and reader friendly manner.

7. Mention statistical analysis techniques in separate section

8. Mention the name of the statistical software used

9. The discussion is very lengthy with many repetitions. Concise the discussion for better readability so that it will be reader friendly.

10. Justify the practical significance of classifying trait ASMR by ASMR-15 tool in real world.

11. Mention the strategies used to control bias

12. Declare any conflict of interest.

---

## [Author Response · Author response to Decision Letter 1]

12 May 2025

Editor comments

• We have revisited these guidelines and made the required edits.

2. Thank you for uploading your study's underlying data set. Unfortunately, the repository you have noted in your Data Availability statement does not qualify as an acceptable data repository according to PLOS's standards. At this time, please upload the minimal data set necessary to replicate your study's findings to a stable, public repository (such as figshare or Dryad) and provide us with the relevant URLs, DOIs, or accession numbers that may be used to access these data. For a list of recommended repositories and additional information on PLOS standards for data deposition, please see https://journals.plos.org/plosone/s/recommended-repositories.

• The data set is now available via the OSF: https://osf.io/j34d8/

• This has been updated accordingly.

4. We note you have included a table to which you do not refer in the text of your manuscript. Please ensure that you refer to Table 1 and 2 in your text; if accepted, production will need this reference to link the reader to the Table.

• Our apologies for this oversight. This has been corrected, see lines 179 and 311.

Reviewer #1:

1. Consider providing a clearer theoretical rationale as to why trait-ASMR should be prioritized over ASMR-sensitivity in explaining individual differences.

• Thank you for this suggestion. We have expanded on this point on lines 94-97 and 107-110.

2. The identification of four trigger categories through PCA is a valuable contribution, but how do these findings align with prior trigger classification frameworks? A more detailed comparison would be beneficial.

• We have expanded in the summary, on lines 564-566, how our findings align with the previous trigger classification studies, which we feel more clearly signposts the reader to the comparisons provided throughout the discussion section. We agree that there needs to be much more detailed consideration of trigger categorisation (theoretical and applied) within the ASMR literature and we hope our findings lay the groundwork for further debate, but that is beyond the scope of this paper. We are also mindful of Reviewer 3’s concerns regarding the discussion being too long.

Reviewer #2: Dear Authors,

This research employs a data-driven approach using PCA, k-means clustering, and MANOVA, which adds valuable insights to the ASMR field. However, several aspects require further clarification and improvement.

Please consider the following modifications (Major Revisions):

1. Justification of Statistical Approaches

While k-means clustering, PCA, and MANOVA are well-suited for your study, please provide a justification for using these methods over alternative clustering or dimensionality reduction techniques such as hierarchical clustering or factor analysis.

• The primary aim of the study was to identify variability within the ASMR-15 subscale profiles that could be contrasted with ASMR-sensitivity, hence using k-means cluster analysis to explore these differences. The goal was not to conceptualise the relationships between the ASMR-15 subscales, and thus hierarchical clustering was judged not to be appropriate. Likewise, we chose PCA as our focus was not on the underlying conceptual relationships between the 58 triggers, but an exploration of trigger categorisation to explain these finding within a real-world context and make the interpretations of the MANOVAs more practical. Again, this meant that our focus was not on the underlying conceptual and statistical relationship between the triggers, and consequently factor analysis was not considered.

Additionally, discuss the assumptions and limitations of PCA in categorizing ASMR triggers, particularly how it handles cross-loading variables and whether alternative factor rotation methods (e.g., oblique rotation) were considered.

• Given the large number of trigger contexts, it would not be entirely unexpected that some triggers with a common label, e.g., Sales Roleplay, could show orthogonal characteristics, but other roleplay triggers e.g. Clinical roleplay show cross-loading. Kim & Mueller (1978) suggested that that in the exploratory stages of analysis an orthogonal rotation may be simpler to understand. Our varimax analysis suggested that items from the same category, e.g., IPC, formed two sub-sets: those with orthogonal characteristics, but also some evidencing cross-loadings. This was therefore not a simple pattern. Our strategy was to remove the cross loadings triggers to make a simplified pattern (Table 4, PCA01) and then subsequently employ those cross-loadings triggers in a second analysis (Table 5, PCA02) By employing the two successive analyses, a greater number of trigger contexts could be explored in the subsequent MANOVAs.

2. Handling of Cross-Loading in PCA

The PCA analyses identified significant cross-loadings, leading to the exclusion of multiple triggers.

Please clarify why a higher threshold (≥0.400) for factor inclusion was chosen and whether alternative methods for handling cross-loadings (e.g., exploratory factor analysis with confirmatory validation) were considered.

• We chose a threshold of ≥0.400 based on Stevens (2002) who made this recommendation as appropriate for interpretative purposes. Considering the extent of the number of triggers that were removed due to this cut-off, we feel that this was a correct approach. We identified contextual differences within the same category membership across the two PCAs (e.g. Roleplay triggers in PCA01 that are situational in nature / Roleplay triggers in PCA02 that are clinical in nature) and which may not have been so readily identified with a softer threshold.

• As we outlined above, our aim was not to explore relationships between triggers, rather create a clearer picture of trigger categorisation that could be more readily applied to real-world ASMR experience, therefore we chose not to use exploratory factor analysis.

3. Effect Size Interpretation and Practical Significance

Given the large sample size (n = 16,679), almost all comparisons were statistically significant. However, the effect sizes (Cohen’s d, η²) are often small to medium, making practical significance unclear. Please emphasize which findings are most meaningful in real-world ASMR categorization rather than focusing solely on statistical significance.

• We have removed wording relating to significance in the discussion unless of particular importance to a finding. Whilst the focus was always on effect sizes, we appreciate that referring to significance in particular where this was accompanied with a small effect size muddied the interpretation. We have retained mention of a non-significant finding as we feel a lack of significance in such a large sample is important and should be emphasised.

4. Potential Confounding Variables

The study does not account for external factors that may influence ASMR perception, such as prior exposure to ASMR, neuroticism, anxiety levels, or sensory processing sensitivity. If these variables were not measured, acknowledge them as limitations.

• Thank you for this suggestion. We have added this to lines 662-664 in the discussion

Trait-ASMR Classification and Cluster Validity

The k-means clustering approach categorized participants into High, Medium, and Low trait-ASMR groups. Please clarify whether alternative clustering solutions (e.g., a four-cluster or two-cluster model) were tested and how the final three-cluster solution was determined as optimal.

• Thank you for raising this important point. We chose a three-cluster solution for our initial investigations as a two-cluster model does not tell us anything more beyond High / Low grouping and does not provide any element of dimensionality in the data. If the observed effect sizes had been greater, we would have considered alternative models to unpick these nuances. However, as we predominantly identified small to medium effect sizes, any increase in the number of clusters would have made the analyses unduly complex to interpret and would not have been any more informative. As such we decided the three-cluster model was optimal for this study.

5. Demographic Influence on ASMR Experience

The study includes a diverse sample but does not analyze whether age, gender, or ethnicity influenced ASMR sensitivity or trait-ASMR classification. Consider running additional subgroup analyses to determine if demographics significantly impact ASMR experience and trigger preference.

• Thank you for this suggestion. We have re-run all analyses controlling for age and gender and which has resulted in minimal change in effect sizes and, for some analyses, no change. We feel ethnicity was not feasible to include as participants could select multiple ethnicities as appropriate, therefore this categorisation would have been extremely nuanced and likely uninformative.

6. Clarification of "Authentic" vs. "Non-Authentic" ASMR

The distinction between "Authentic ASMR" (tingles originating in the head/neck) and "Non-Authentic ASMR" (tingles elsewhere) is interesting but may not fully capture the subjective variability of ASMR experiences. Please discuss whether alternative ASMR classifications (e.g., intensity-based or multidimensional grouping) were considered.

• We absolutely agree that the aASMR / nASMR categorisation does not capture variability of ASMR experience and our findings speak clearly to this point. Our aim was to compare this binary categorisation which dominates the ASMR literature with a data-driven approach using the ASMR-15 as this tool is becoming more widely used within research. However, there are indeed many potential multi-dimensional groupings that may predict ASMR-experience (e.g., personality traits / sensory processing sensitivity / neural activity) which have more typically been investigated as outcome variables rather than predictor variables. We welcome future research that takes on wider dimensional rather than categorical approaches as this investigation is still lacking within the ASMR literature.

7.Consider discussing any psychometric limitations (e.g., response bias, scale ceiling effects).

• We agree that there are important issues and have addressed these on lines 672-676.

8. Discussion of Ethical Considerations

Since the study recruited participants through social media platforms, please briefly mention any ethical considerations (e.g., potential biases in self-selection, data privacy concerns).

• Thank you for this suggestion. We have acknowledged biases in self-selection in lines 661-662 and 666-669. With regard to data privacy, the survey was anonymous, and no data was contributed by the participants that allowed any form of identification. This has been clarified on lines 235-237.

Reviewer #3: Comments/queries

1. Kindly make a structured abstract

• Whilst we have no objection to a structured abstract, PLoS ONE formatting guidelines do not specify this and we have been instructed by the handling editor to ensure our manuscript follows these.

2. In all the figures both mean and median scores are evaluated and incorporated in the same image.

Explain why it is done.

• The software programme that was used to generate these figures creates the median as default and with the option to also include the mean. We felt inclusion of these data enabled full transparency of the findings.

3. Expand the abbreviations wherever it is used initially

• Thank you for pointing out this oversight and we have clarified this where missing in the manuscript. We have also removed the term IV (referring to independent variable) on lines 437, 445, 458, and 467 as this was included in error.

4. Kindly attach the ASMR-15 tool in the appendix.

• The ASMR-15 is available in the full survey which is available via the OSF: https://osf.io/j34d8/

5. If permission is obtained to use ASMR-15 tool, kindly mention it

• We have ensured that the authors have been cited accordingly.

6. The Methodology section of the manuscript require reorganization of contents. Draft it in a sequential and reader friendly manner.

• The method has been restructured by removing repetition regarding participants’ demographics on lines 186-189 and moving the content regarding the ASMR-15 prior to the detail regarding tingle location and intensity. This now follows a more logical structure that is replicated in the results and the discussion.

7. Mention statistical analysis techniques in separate section

• We have renamed ‘Treatment of data’ section in the Methods to ‘Statistical analyses’.

8. Mention the name of the statistical software used

• This detail has been included on line 241.

9. The discussion is very lengthy with many repetitions. Concise the discussion for better readability so that it will be reader friendly.

• Thank you for pointing this out. We have revisited the discussion section and removed repetitions / made edits which we feel makes this more reader friendly.

10. Justify the practical significance of classifying trait ASMR by ASMR-15 tool in real world.

• Thank you for this suggestion. This aligns with comments from Reviewer 1 who posed a similar question, and we have added further text to the introduction (lines 94-97 and 107-110) to emphasise why trait-ASMR should be considered an individual difference. The differing ASMR-15 profiles we observed across the trait-ASMR groups provides support for this, and we hope, with the advised edits to the discussion, this reads more readily. On lines 664-667 we have raised potential issues with the ASMR-15 especially within a targeted sample.

11. Mention the strategies used to control bias

• There are many potential biases in ASMR research and which need to be considered within the context of the specific research focus. We did not account for any in this study, however, as our investigations were not specific to other personality characteristics. We were investigating trait-ASMR characteristics from a baseline perspective which can be used to guide future research investigating the real-world impact of ASMR propensity. We have acknowledged potential confounds in the discussion on lines 662-664 and which align with similar comments raised by Reviewer 2.

12. Declare any conflict of interest.

• We have included a statement on the title page that there are no competing interests.

---

## [Editor Report · Decision Letter 1]

Do whispering minds tingle alike? Exploring the relationship between ASMR-sensitivity, trait-ASMR, and trigger preference

PONE-D-25-05327R1

Dear Dr. Greer,

We’re pleased to inform you that your manuscript has been judged scientifically suitable for publication and will be formally accepted for publication once it meets all outstanding technical requirements.

Kind regards,

PUGAZHENTHAN THANGARAJU, M.D.,Ph.D., FRCP (LONDON)., FRCP (GLASGOW).,MBA.,

Academic Editor

PLOS ONE
---

## [Editor Report · Acceptance letter]

PONE-D-25-05327R1

PLOS ONE

Dear Dr. Greer,

I'm pleased to inform you that your manuscript has been deemed suitable for publication in PLOS ONE. Congratulations! Your manuscript is now being handed over to our production team.

Kind regards,

on behalf of

DR. PUGAZHENTHAN THANGARAJU

Academic Editor

PLOS ONE